# A Bioavailable Strontium Isoscape of Australia

Anthony Dosseto[1], Florian Dux[1], Clément Bataille[2,3,] Patrice de Caritat[4,5]

[1]Wollongong Isotope Geochronology Laboratory, Environmental Futures, School of Science, University of Wollongong, Wollongong NSW 2522, Australia
[2]University of Ottawa, Earth and Environmental Sciences, Ottawa, Canada
3 Purdue University, Forestry and Natural Resources, West Lafayette, Indiana, USA
[4]Geoscience Australia, GPO Box 378, Canberra ACT 2601, Australia
[5]John de Laeter Centre, Curtin University, Bentley WA 6102, Australia

*Correspondence to*: Anthony Dosseto (tonyd@uow.edu.au)

## Abstract

Strontium isotope ratios ($^{87}Sr/^{86}Sr$) at the Earth's surface offer powerful tools for geological, environmental, and archaeological applications. In minerals and biological materials, $^{87}Sr/^{86}Sr$ reflects the isoztopic composition of the local bedrock and derived soils. In Australia, however, large regional-scale surveys of bioavailable $^{87}Sr/^{86}Sr$ remain scarce. Here, we present a new dataset of bioavailable $^{87}Sr/^{86}Sr$ ratios from 278 catchment outlet (floodplain) sediment samples, spanning inland southeastern Australia (South Australia, New South Wales, Victoria), northern Western Australia, the Northern Territory, Queensland (north of 21.5°S), and the Yilgarn Craton in southern Western Australia. Combined with more than 20,000 global Sr isotope measurements, this dataset was used to generate a high-resolution isoscape of Australia using a well-established random forest spatial regression framework (Bataille et al., 2020).

Australian bioavailable $^{87}Sr/^{86}Sr$ values span a narrower range (0.70501–0.78121) compared to co-located bulk sediment values (0.70480–1.09089) (Caritat et al., 2022, 2023, 2025b), reflecting the influence of soluble and exchangeable mineral phases and atmospheric inputs such as rain and dust/seaspray. The predicted isoscape reproduces major geological patterns, with higher values over ancient crustal provinces like the Yilgarn Craton and eastern Palaeozoic orogens, and lower values across younger sedimentary basins and coastal margins. Model uncertainty, assessed via quantile random forest regression, is lowest across well-sampled, geologically stable regions where the model is well-trained and highest in poorly-sampled regions and lithologically complex zones. Despite remaining spatial gaps and areas of high prediction uncertainty, our model offers significantly improved coverage and resolution for Australia compared to other global or regional isoscapes. It also provides a scalable framework for updating the Australian isoscape as sampling density increases. This isoscape establishes a robust baseline for applications in provenance research, palaeoecology, and environmental geochemistry.

# 1 Introduction

Provenancing—the ability to trace the geographic origin of materials—is a vital tool across disciplines such as ecology, archaeology, food authentication, and forensic science (Hobson et al., 2010; Bentley, 2006; Kelly et al., 2005; Voerkelius et al., 2010; Meier-Augenstein, 2017). In Australia, this capability is particularly valuable due to the continent's unique biogeography, ancient human history, and significant food production systems. Australia's ecosystems have evolved in isolation, resulting in many endemic species with region-specific foraging or migration behaviours (Crisp and Cook, 2013). Understanding the ecological histories of both extant and extinct fauna requires geochemical tools capable of linking individuals to specific landscapes (Hobson et al., 2010; Bataille et al., 2020). Similarly, tracing the origin and movement of archaeological materials and human remains is fundamental to understanding the peopling of Australia, the development of long-distance trade routes, and the complex land-use practices of Aboriginal peoples (Malaspinas et al., 2016; Clarkson et al., 2017). In contemporary applications, provenance tools are increasingly used to verify the geographic origin of agricultural commodities - such as wine, seafood, and grain (Kelly et al., 2005; Almeida and Vasconcelos, 2003) - which is important both for biosecurity and for protecting the reputation of high-value Australian exports. Despite these diverse applications, there remains a critical limitation: Australia lacks a continent-scale framework for biologically relevant isotopic provenancing.

The strontium (Sr) isotope ratio ($^{87}Sr/^{86}Sr$) is a powerful tool used across a wide range of disciplines, including geoscience, palaeoecology, archaeology, and forensic science. In geoscience, $^{87}Sr/^{86}Sr$ ratios are applied to investigate processes such as continental weathering, sediment provenance, and the evolution of crustal materials over geological time (e.g. Bataille et al., 2020; Mcnutt, 2000). In archaeology and forensic science, variations in $^{87}Sr/^{86}Sr$ have been used to trace the origin of artefacts, human remains, and food products by linking them to specific geological regions (e.g. Frei and Frei, 2013; Willmes et al., 2014; Voerkelius et al., 2010). This broad applicability stems from the fact that $^{87}Sr/^{86}Sr$ ratios remain stable during biological and chemical processes, preserving the isotopic signature of the original environment (Gosz et al., 1983; Nebel and Stammeier, 2018).

The $^{87}Sr/^{86}Sr$ ratio of a material varies across the landscape due to differences in the age and Rb/Sr ratio of underlying rocks. Older rocks, particularly with a high Rb/Sr, such as granites and metamorphic rocks, tend to have higher $^{87}Sr/^{86}Sr$ ratios, whereas younger volcanic rocks and carbonates typically display lower values (e.g. Bataille et al., 2020; Mcnutt, 2000). Weathering of these rocks releases Sr into soils, waters, and vegetation, transferring the distinct isotopic signatures to the biosphere. This natural variation enables researchers to map $^{87}Sr/^{86}Sr$ distributions across regions ("isoscapes") and use these maps to infer the provenance of materials, reconstruct past human and animal movements, and investigate environmental and geological processes (e.g. Bataille et al., 2018; Caritat et al., 2023).

When using Sr isotopes as a provenance tool, a well-constrained Sr isoscape is essential (Hobson et al., 2010). Although a few efforts have been made to develop Sr isoscapes in Australia, existing datasets remain limited in spatial coverage or relevance for biological provenancing. The study by Adams et al. (2019) focused on the Cape York Peninsula and represents a foundational effort in mapping bioavailable $^{87}Sr/^{86}Sr$ for archaeological applications. Rippon et al. (2020) further demonstrated

the use of Sr isotopes in an Australian archaeological context, though at a more site-specific scale. Caritat et al. (2022, 2023, 2025b) provide extensive $^{87}Sr/^{86}Sr$ data for bulk sediment across inland southeastern, northern Australia, and the Yilgarn Craton, respectively. However, provenance studies in archaeology, forensic science, and palaeoecology are based on comparing biological tissues of unknown origin with the isotopic composition of bioavailable Sr—the fraction accessible to plants and animals—rather than bulk or total sediment Sr (Bataille et al., 2020; Capo et al., 1998). Bioavailable Sr more accurately reflects the isotopic signature incorporated into biological tissues, making it critical for robust provenance interpretations in these fields.

The aims of this study are (1) to present a new bioavailable Sr isotope dataset from three large regions in Australia and (2) to develop the first continent-scale isoscape of bioavailable $^{87}Sr/^{86}Sr$ for Australia. The new dataset of bioavailable Sr isotope ratios is based on 278 catchment outlet sediment samples, spanning a range of climatic zones and geological provinces across the continent. These data were integrated with a global compilation of >20,000 plant, soil, and water Sr isotope ratios—including an additional 292 georeferenced samples from Australia—to train a random forest regression model, enabling spatial prediction of bioavailable $^{87}Sr/^{86}Sr$ values across unsampled regions. The resulting Sr isoscape provides a preliminary, geochemically informed, spatially continuous framework to support provenance research in archaeological, ecological, and forensic contexts, and could be advantageously used in combination with the recently presented national-scale lead (Pb) isoscape for Australia (Desem et al., 2025). By addressing the current lack of bioavailable Sr data for Australia, this study enhances the application of isotope-based provenancing across diverse environmental and cultural landscapes.

**2 Study Area**

This study focuses on terrestrial samples collected across northern, southeastern, and southwestern Australia as part of the National Geochemical Survey of Australia (NGSA) project, a continental-scale geochemical mapping program that systematically sampled catchment outlet sediments across approximately 80% of the Australian landmass (Caritat and Cooper, 2011; Caritat and Cooper, 2016; Caritat et al., 2022). The selected catchments vary widely in climate, geology, and landscape history, spanning tropical, semi-arid, and temperate zones.

In northern Australia, catchments are dominated by deeply weathered Precambrian bedrock, including granites, gneisses, and Proterozoic sedimentary rocks, overprinted by extensive regolith development (Caritat et al., 2023). Southeastern Australia features a mix of Palaeozoic granites, volcanic provinces, and younger sedimentary basins (Caritat et al., 2022). Western Australia, particularly the Yilgarn Craton, represents some of the oldest continental crust on Earth, consisting mainly of Archean granite-greenstone terranes and Proterozoic basins (Caritat et al., 2025b).

The sampled areas are generally characterised by minimal recent glaciation or rejuvenation, resulting in thick, stable weathering profiles (Wilford, 2012). In many locations, aeolian processes contribute to sediment mixing, particularly in arid and semi-arid regions (Caritat and Cooper, 2011). Catchment outlet sediments therefore integrate signals from diverse lithological sources, modified by long-term weathering and surface processes.

Bioavailable Sr isotope ratios ($^{87}$Sr/$^{86}$Sr) were measured on the <2 mm fraction of these sediments to characterise the isotopic landscape ("isoscape") at a continental scale. While sampling density (~1 site per 5,200 km²) limits fine-scale spatial resolution, the large coverage provides a robust first-order framework for provenance and environmental studies across much of Australia.

## 3 Material and methods

### 3.1 Material

This study makes use of archived sediment samples collected by the NGSA. The NGSA targeted fine-grained fluvial and alluvial sediments at major catchment outlets, which serve as effective integrators of upstream geological inputs through natural weathering and sediment transport. Sampling was conducted at an ultralow density of approximately one site per 5,200 km², designed to capture large-scale geochemical variation across diverse climatic and geological regions. This approach follows protocols adopted in other national-scale surveys (e.g. Ottesen et al., 1989; Bølviken et al., 2004).

For the present study, we selected 278 NGSA samples from across northern, southeastern, and southwestern Australia (Figure 1), prioritising areas not previously covered by bioavailable Sr isotope analysis, but previously studied for bulk Sr isotopes. These samples underpin the development of the first continental-scale isoscape of bioavailable $^{87}$Sr/$^{86}$Sr in Australia. The catchment outlet sediments share characteristics with floodplain deposits, being deposited during receding floodwaters, but also reflect aeolian influences in some regions. The sampled settings were typically vegetated and biologically active, with soils forming on transported alluvial parent material.

In contrast to the bulk Sr isotope ratios reported in Caritat et al. (2022, 2023, 2025b), which were measured on milled and fully digested <2 mm fraction of 'bottom outlet sediment' (BOS) samples (average depth from 0.6 to 0.8 m), the bioavailable Sr in this study was extracted from the <2 mm fraction of the overlying 'top outlet sediment' (TOS) samples (which interfaces with the biosphere) using mild ammonium acetate solution to isolate the exchangeable and soluble fraction relevant to biological uptake. The TOS samples were taken from shallow pits at ~0.1 m depth (Lech et al., 2007). In the laboratory, samples were air-dried, milled, and processed following documented NGSA protocols (Caritat et al., 2010). While the low sampling density constrains fine-scale resolution, the broad spatial coverage provides a valuable baseline for Sr isotopic variation and supports future provenance, palaeoenvironmental, and landscape evolution studies.

### 3.2 Methods

Samples were prepared and analysed for $^{87}$Sr/$^{86}$Sr ratios at the Wollongong Isotope Geochronology Laboratory (WIGL). Bioavailable Sr was extracted from the <2 mm fraction of sediment samples using a 1 M ammonium acetate (NH$_4$OAc) leach at pH 7, targeting the exchangeable and readily soluble Sr fraction (Moffat et al., 2020). Approximately 1 g of dried, milled

sample was weighed into a 15 mL polypropylene tube and leached with 2.5 mL of the NH$_4$OAc solution for 24 hours on a table shaker at 3000 rpm. Following leaching, the supernatant was filtered through a 0.45 μm PTFE syringe filter to remove particulates and organic material.

The filtered leachate was evaporated to incipient dryness on a hotplate at 100 °C, then re-dissolved in 2 mL of 2 M HNO$_3$. A 1:100 dilution in 0.3 M HNO$_3$ was prepared from each sample to screen for Sr concentration prior to chromatographic separation. Strontium was separated from the sample matrix using an automated low-pressure chromatographic system (Elemental Scientific prepFAST-MC™) equipped with 1 mL Sr–Ca resin columns (Eichrom™), following the procedure of Romaniello et al. (2015). Purified Sr fractions were collected and re-dissolved in 0.3 M HNO$_3$ prior to isotope ratio analysis.

Strontium isotope ratios were measured using a Thermo Scientific Neptune Plus multi-collector inductively coupled plasma mass spectrometer (MC-ICP-MS) at WIGL. Samples were introduced via an ESI Apex-ST PFA MicroFlow nebuliser (~0.1 mL min$^{-1}$ uptake rate) coupled to an SSI quartz dual cyclonic spray chamber, with a jet sample cone and X-skimmer cone configuration. Analyses were performed in low-resolution mode. Instrument tuning was conducted at the start of each analytical session using a 20 ppb Sr solution, with typical $^{88}$Sr signal intensities of ~4 V. Isotopes $^{88}$Sr, $^{87}$Sr, $^{86}$Sr, $^{85}$Rb, $^{84}$Sr, and $^{83}$Kr were collected simultaneously on Faraday detectors. Instrumental mass bias was corrected using internal normalisation of $^{87}$Sr/$^{86}$Sr to the known $^{88}$Sr/$^{86}$Sr ratio via the exponential law. Isobaric interferences from $^{87}$Rb and $^{86}$Kr were corrected using intensities measured at masses 85 and 83, respectively.

The National Institute of Standards and Technology (NIST) strontium carbonate isotope Standard Reference Material SRM987 was used as a secondary standard and analysed after every five samples to monitor instrument stability and analytical accuracy. The average $^{87}$Sr/$^{86}$Sr ratio measured for SRM987 over the analytical campaign was 0.709810 ± 0.000044 (2SE, n = 90), which is notably lower than the certified value of 0.710252 ± 0.000013 reported by Weis et al. (2006). This offset is likely due to long-term instrument drift or detector ageing. To correct for this, $^{87}$Sr/$^{86}$Sr ratios measured in samples were normalised session-by-session to the mean SRM987 value obtained during that session, using the Weis et al. (2006) value as a reference. This approach ensures data comparability across sections and facilitates future re-referencing. Full procedural accuracy—encompassing leaching, filtration, and chromatographic separation—could not be assessed due to the lack of an appropriate certified reference material that undergoes the same sample preparation steps as the sediment leachates. No standard reference material currently exists that replicates the matrix and leaching behaviour of ammonium acetate-extractable bioavailable Sr. As such, while instrumental accuracy is constrained by SRM987, the potential for matrix-specific biases or fractionation during leaching could not be independently verified.

To assess full procedural reproducibility, a total of ten field duplicates were independently leached, processed, and analysed for $^{87}$Sr/$^{86}$Sr. These replicate pairs span the full range of observed values in the dataset. Paired results show excellent reproducibility, with differences between replicates ranging from ±0.00003 to ±0.00015, and an average absolute difference of 0.000043 (median = 0.000030). These results are consistent with those reported in similar studies using NH$_4$OAc extraction and MC-ICP-MS analysis (e.g. Moffat et al., 2020). Precision based on 2 standard errors ranged from ±0.000007 to ±0.000036 across replicates, reflecting both analytical performance and micro-scale sediment heterogeneity. Total procedural blanks were

low, ranging from 0.07 to 0.26 ng Sr (n = 8), and are negligible relative to sample Sr concentrations. Overall, the quality of the $^{87}$Sr/$^{86}$Sr dataset is considered appropriate for regional-scale isoscape modelling. Given the level of analytical precision observed, we report $^{87}$Sr/$^{86}$Sr values to the fifth decimal place. Replicate measurements show differences within acceptable limits, and no significant additional variation is observed among field duplicates beyond that attributable to sample heterogeneity and standard analytical uncertainty.

### 3.3 Strontium Isoscape Calculation

The Sr isoscape was generated using the random forest regression model approach described by Bataille et al. (2020). Like many machine-learning approaches, random forest regression performs best when trained on large, diverse datasets, particularly when incorporating numerous environmental predictors. Bataille et al. (2020)emphasize that data-rich regions such as Europe provide optimal conditions for training accurate and unbiased models. In contrast, data-poor regions pose challenges for testing model performance and for model generalization. When local datasets are small or spatially biased towards specific environments, locally-trained models may exhibit inflated performance metrics and poor extrapolation beyond the training range. To mitigate these risks, Bataille et al. (2020) recommend using global or extended bioavailable datasets to train regional isoscapes in such contexts. This strategy enhances model generalisability and reduces overfitting, especially in geologically complex or under-sampled areas. Following this approach, we trained the Australia-wide model using a global bioavailable $^{87}$Sr/$^{86}$Sr dataset comprising over 20,000 measurements from plant tissues, soil exchangeable fractions, and water samples, which includes 332 georeferenced Australian samples (Crook et al., 2017; Adams et al., 2019; Raiber et al., 2009; Palmer and Edmond, 1989; Goldstein and Jacobsen, 1987). This dataset was augmented with the 278 new bioavailable $^{87}$Sr/$^{86}$Sr measurements from Australian sediment leachates reported in this study, together with 90 unpublished georeferenced plant samples (Dosseto, unpub.), thereby substantially expanding the empirical representation of Australian environments in the model. To provide transparency on the training dataset, we include in the Supplementary Information (Table S1) a detailed breakdown of the sample types used. The compilation comprises 23.6% plant samples, 16.1% soil samples, and 25.7% water samples, with the remaining 34.6% representing other categories (e.g., animal tissues, shells, and rock). While Australia data remains limited and spatially biased, the combined dataset captures a wide range of geological and environmental conditions across Australia, improving the robustness and transferability of the model across the continent. This approach provides a strong foundation for continental-scale predictions, however, a dedicated Australia-specific random forest model should be developed in the future as sampling density and geographic coverage increase across the continent. This regionally-calibrated model will provide predictions tailored to local geological and environmental conditions and could leverage the wealth of existing Australia geophysical covariates.

To train the model, we leverage 26 geospatial variables representing geological, climatic, soil, hydrological, and vegetation attributes with global scale extent, prepared and organized by Bataille et al. (2020) (see Table 1). This integration of Sr isotope observations with environmental covariates enables the prediction of bioavailable $^{87}$Sr/$^{86}$Sr values across unsampled regions,

producing a continuous spatial model of Sr isotopic variation at a resolution of approximately 0.012° × 0.012° (~1 km × 1 km).

In addition to mean predicted bioavailable $^{87}Sr/^{86}Sr$, we calculated a spatially-explicit uncertainty layer using quantile regression random forest to calculate a pseudo standard deviation (SD) as proposed by Funck et al. (2020). This uncertainty layer quantifies the uncertainty of $^{87}Sr/^{86}Sr$ predictions across the landscape. Areas with low SD values indicate high model confidence—often reflecting well-characterised environmental conditions or strong covariate signal—while higher SD values highlight regions where predictions are more uncertain due to environmental and geological complexity along with limited training data.

## 4 Results

### 4.1 Bioavailable and Bulk $^{87}Sr/^{86}Sr$ Distributions

Bioavailable $^{87}Sr/^{86}Sr$ ratios in the Australian sediment samples range from 0.70501 to 0.78121, with a mean of 0.72131 (Caritat et al., 2025a). Distinct regional patterns are evident (Figure 2; Table 2). Samples from southeastern Australia (South Australia, New South Wales, Victoria) display the lowest mean $^{87}Sr/^{86}Sr$ (0.71277), with values ranging from 0.70739 to 0.71908. In contrast, samples from northern Australia (Northern Territory, northern Western Australia, and Queensland north of 21.5°S) exhibit a higher mean of 0.72150 and a broader isotopic range (0.70501–0.78121), reflecting greater geological complexity diversity and complexity. Sediments from southwestern Australia (Yilgarn Craton) are more radiogenic, with a mean $^{87}Sr/^{86}Sr$ of 0.72468 and values ranging from 0.71153 to 0.75274.

In comparison, bulk sediment $^{87}Sr/^{86}Sr$ values measured on co-located (but deeper), and fully digested samples, are systematically higher and more variable, spanning from 0.7048 to 1.0909 (mean = 0.7501) (Caritat et al., 2025b). Across all regions, bioavailable Sr is consistently less radiogenic than bulk Sr (Figure 3; Figure 4). The average offset between bioavailable and co-located bulk $^{87}Sr/^{86}Sr$ values is approximately:

- 0.010 in southeastern Australia,
- 0.028 in northern Australia,
- 0.040 in southwestern Australia.

This systematic difference reflects different geochemical behaviour of Sr during weathering processes. The broader range in bulk sediments results from the inclusion of minerals with a large range of rubidium/strontium ratio leading to distinct $^{87}Sr/^{86}Sr$ ratios in the same rock usually more radiogenic for micas and less radiogenic for feldspars. In contrast, the bioavailable fraction displays usually a lower range and values because it comes primarily from the weathering of less radiogenic Sr-rich mineral such as feldspars or carbonates. In addition, bioavailable Sr is buffered by the addition of less radiogenic and Sr-rich soluble phases and atmospheric inputs such as rainwater, sea salt and dust—exhibits a narrower range. To summarise, these differences are not attributable to sample size or biases (n = 576 for bulk, n = 278 for bioavailable), but reflect the differential weathering

rates and contributions of underlying bedrock minerals and the addition of other Sr sources to the bioavailable pool (see review by Capo et al., 1998). The distributions are summarised in Figure 3, and individual paired comparisons are shown in Figure 4.

## 4.2 Predicted Isoscape and Regional Patterns

The bioavailable $^{87}Sr/^{86}Sr$ isoscape of Australia was generated using a random forest regression model trained on the 278 sediment samples from this study and supplemented by over 20,000 global Sr isotope values (see Section 3.3). The resulting model predicts broad-scale isotopic variation that aligns with major geological provinces (Figure 5; Supplementary Figure S1). Elevated $^{87}Sr/^{86}Sr$ values are predicted over the Yilgarn Craton and parts of eastern Australia, consistent with exposure of ancient Archean and Palaeozoic bedrock in this region. In contrast, younger sedimentary basins of central and northern Australia show lower predicted values. Coastal areas, particularly in northern and northwestern Australia, also exhibit low $^{87}Sr/^{86}Sr$ values, likely influenced by sea salt aerosols and more recent marine sediments deposited from recent eustatic sea level changes. These regional patterns mirror Australia's lithological and climatic gradients and confirm the strong geological control on the bioavailable Sr pool and their usefulness for many provenance studies in the region.

## 4.3 Prediction Uncertainty

A map of pseudo standard deviation (SD) values was generated to estimate model uncertainty (Figure 6; Supplementary Figure S2). Prediction uncertainty is lowest (SD < 0.003) across much of southeastern Australia and parts of the interior, reflecting areas where the model is well-constrained by training data and geologically consistent covariates (e.g. bedrock $^{87}Sr/^{86}Sr$, dust inputs). In contrast, higher uncertainty (SD > 0.005) is observed across large parts of northern Australia, the southwestern margin of the Yilgarn Craton, and some inland areas of South Australia and the Northern Territory. These regions are characterised by geological heterogeneity, variable environmental conditions, or sparse sample coverage. This SD map should serve as an important tool for interpreting prediction reliability and for guiding future sampling strategies.

## 4.4 Model Performance and Variable Importance

Cross-validation of the global random forest model yielded an $R^2$ of 0.590, RMSE of 0.00933, and MAE of 0.00278, reflecting moderate predictive performance across diverse global environments. While these values reflect model fit at the global scale, regional performance within Australia may differ depending on sampling density and local geological complexity. Variable importance analysis using the VSURF (Variable Selection Using Random Forests) algorithm (Genuer et al., 2010) identified ten key predictors influencing the global model of bioavailable $^{87}Sr/^{86}Sr$: the 25[th] percentile of predicted bedrock $^{87}Sr/^{86}Sr$ values (srsrq1), maximum geological age (maxage_geol), simulated distance to dust/seaspray sources or coastline (distance), mean terrane age (age), mean annual precipitation (map), Bouguer gravity anomaly (bouger), black carbon deposition from fossil fuels (foss), dust deposition flux (dust), mean annual temperature (mat), and the aridity index (ai). These variables were selected based on their contribution to reducing prediction error across the global training dataset. Together, they highlight the

dominant influence of geological composition and crustal evolution on bioavailable Sr isotope ratios, as well as the modifying effects of atmospheric deposition and climatic conditions. Although the model was globally trained, the inclusion of variables

such as dust, aridity, and fossil fuel-derived particulates is consistent with known Sr cycling mechanisms in Australia, where aeolian transport, climatic gradients, and anthropogenic inputs significantly influence the mobile Sr pool.

## 5 Discussion

This study presents a new machine learning-based isoscape of bioavailable $^{87}Sr/^{86}Sr$ for Australia, generated using a random forest model trained on 278 sediment samples and supplemented with over 20,000 global Sr isotope data points, including

Australian bioavailable Sr isotope data from Adams et al. (2019). The resulting isoscape represents the most comprehensive spatial prediction of bioavailable Sr isotopic variation to date for the continent, with broad implications for geochemical, archaeological, ecological, and forensic applications. The spatial distribution of predicted $^{87}Sr/^{86}Sr$ values closely reflects the geological architecture of the Australian continent. Higher ratios correspond with older crustal provinces, notably the Archean Yilgarn Craton and Palaeozoic orogens in eastern Australia, while lower ratios are associated with younger sedimentary basins,

coastal plains, and regions influenced by marine or aeolian inputs. These patterns are consistent with established geochemical principles whereby radiogenic $^{87}Sr$ accumulates over time in lithologies with high Rb/Sr ratios such as granites and gneisses (Faure and Powell, 2012), and less radiogenic values are found in mafic, carbonate, or younger siliciclastic environments (Bataille et al., 2020; Willmes et al., 2018).

The key predictors identified by the VSURF analysis—median bedrock $^{87}Sr/^{86}Sr$, geological age, simulated dust source

distance, mean terrane age, and mean annual precipitation—reflect globally important controls on bioavailable Sr isotope variation. The prominence of geological variables is consistent with well-established geochemical principles: older lithologies with higher Rb/Sr ratios typically contribute more radiogenic $^{87}Sr/^{86}Sr$ signatures to soils and sediments. The inclusion of atmospheric and climatic variables such as dust/seaspray transport distance and precipitation highlights the broader environmental processes that can influence the mobile Sr pool, particularly through aeolian and hydrological redistribution.

While the model was trained on a global dataset, these results are geochemically plausible in the Australian context, where low-relief, deeply weathered landscapes and dust transport are key features (Wilford, 2012; Caritat et al., 2022). The findings underscore the importance of incorporating both geological and surface process variables when predicting bioavailable Sr, especially in regions with extensive regolith cover or where external inputs may obscure local bedrock signals (Bataille & Bowen, 2012).

Prediction uncertainty, visualised through the SD map (Figure 6 and Supplementary Figure S2), is lowest across geologically uniform and well-constrained regions, particularly in southeastern Australia. In contrast, higher uncertainties (SD > 0.005) occur across parts of northern Australia, the southwestern margins of the Yilgarn Craton, and central inland areas, likely reflecting greater geological heterogeneity, limited training data, or environmental complexity. Following recommendations from Willmes et al. (2018) and Scaffidi and Knudson (2020), we emphasise that users of the isoscape should carefully consider

local uncertainty when applying it to provenance assignments or ecological interpretations. Spatially variable prediction confidence is particularly important in forensic and archaeological contexts, where assignment errors can lead to misinterpretation of individual mobility or material origin. Although quantile random forest provides a robust framework for estimating prediction uncertainty, it does not account for spatial autocorrelation. As a result, uncertainty may be overestimated in data-rich regions and underestimated in data-poor areas, especially where covariates are spatially clustered or unevenly distributed. Current advances in machine-learning are attempting to solve these issues to produce more robust spatially-explicit uncertainty estimates.

Compared to existing Sr isoscapes, this study substantially improves both the spatial coverage and resolution of bioavailable $^{87}Sr/^{86}Sr$ predictions for Australia. The regional isoscape developed by Adams et al. (2019) for the Cape York Peninsula represented a foundational effort in biologically relevant Sr modelling in Australia, based on direct measurements from plant, water, and soil leachates. Similarly, Rippon et al. (2020) generated a detailed bioavailable Sr isotope baseline for the Adelaide region using low-mobility fauna, demonstrating the potential for fine-scale Sr-based provenancing in southern Australia. However, the geographic scope of these studies is limited, highlighting the need for a continent-wide framework. The high-resolution isoscape for France by Bataille et al. (2018) demonstrated the value of integrating geological and environmental covariates in a spatial modelling framework, but was developed and calibrated for a European context. The global isoscape by Bataille et al. (2020) provided valuable continental-scale predictions, yet relied on sparse empirical data from Australia and extensive extrapolation over unsampled areas.

The present study addresses this gap by generating the first continent-wide bioavailable $^{87}Sr/^{86}Sr$ isoscape for Australia, based on 278 new sediment leachate measurements integrated into a globally trained machine learning model. This approach increases the empirical representation of Australian geochemical landscapes and provides a robust spatial baseline for provenance, ecological, and archaeological applications. At a regional level, our predicted values for Cape York are consistent with spatial trends reported by Adams et al. (2019), including lower values over western sediments and higher values associated with older basement lithologies in the east. Our predicted values are on average 0.00124 lower (range from 0.03976 lower to 0.03006 higher) than the measured values reported by Adams et al. (2019) (Figure 7). This difference is not statistically significant (Welch's t-test, p = 0.25), and the effect size is small (Cohen's d = 0.19), suggesting that while the model underpredicts slightly, it broadly captures the measured isotopic range. Minor offsets may reflect differences in sampling media (sediment leachates versus in situ biological samples) and the spatial smoothing inherent to machine learning predictions.

Despite its advances, the model has important limitations that users should consider when applying the isoscape for provenance or ecological questions. An important methodological consideration in this study is the use of catchment outlet sediments as the sampling medium for bioavailable Sr. These sediments integrate weathering products from upstream lithologies, offering a spatially averaged signal that reflects regional geochemical variation. While this approach limits the resolution of local-scale isotopic heterogeneity – particularly in geologically complex or transitional zones – it provides a robust first-order approximation of bioavailable $^{87}Sr/^{86}Sr$ at the landscape scale. This is well-suited to the development of continental-scale isoscapes. However, for applications requiring fine-scale provenancing, such as intra-site archaeological mobility or ecological

habitat use, future work should prioritise the incorporation of point-sourced biological samples such as plants, water, or faunal tissues. Expanding such datasets would enhance the spatial resolution and local accuracy of bioavailable Sr predictions across Australia.

A further consideration is the comparability of different sample types (plants, soils, and waters) to the tissues of animals and humans, which are the ultimate focus of most provenance studies. Plant samples provide a direct measure of the Sr isotope composition entering the food chain, as Sr is incorporated into leaves and roots via uptake from soils and waters (Capo et al., 1998; Evans et al., 2006). Soil leachates, such as those analysed in this study, represent the exchangeable and soluble Sr fraction that is available for plant uptake, and thus provide an integrative proxy that is well-suited for large-scale surveys (Moffat et al., 2020; Frei and Frei, 2013). Water samples, by contrast, reflect dissolved Sr inputs from bedrock weathering and atmospheric sources, and are particularly relevant in aquatic ecosystems or for populations that rely heavily on local drinking water (Voerkelius et al., 2010). While each medium has strengths and limitations, all broadly capture the local bioavailable Sr pool and are therefore comparable to the isotopic signatures archived in animal and human tissues such as bone, dentine, enamel, or hair (Bentley, 2006; Copeland et al., 2016). Differences among media should be considered when designing provenance studies, but in general, convergent patterns across plants, soils, and waters provide confidence that these proxies faithfully reflect the Sr available to biological systems.

The model was trained globally, and although it performs moderately well at the global scale ($R^2 = 0.590$, RMSE = 0.00933), the regional accuracy within Australia was not fully tested and may vary significantly, particularly in areas with sparse sampling or complex geology. The spatial resolution is constrained by the original NGSA sampling density (~1 site per 5,200 km$^2$), and several regions, including the interior of northern Australia, Tasmania, and remote coastal zones, remain severely underrepresented. This along with sampling bias towards certain geological settings may limit the generalisability of predictions across unsampled environments.

Critically, users should avoid applying the isoscape in regions where predictor values fall outside the range of the training data, as extrapolation in these areas may yield unreliable results. The current model should be viewed as a foundational framework, not a definitive tool, and its utility and accuracy will improve as more data become available. Expanding bioavailable Sr data collection, including plant, small animals and soil samples, across underrepresented regions of Australia is essential to enhance model accuracy. In particular full use of the remaining NGSA sample set along with targeted sampling in geologically diverse areas coupled with use of Australia-specific geophysical predictors (e.g., radiometric surveys) will be key to develop an increasingly accurate continental scale. This foundational study, however, demonstrates the strong promises of applying strontium isotopes for provenancing Australian animals and materials in the present and in the past.

## 6 Conclusions

This study presents the first machine learning-based isoscape of bioavailable strontium isotope ($^{87}$Sr/$^{86}$Sr) ratios across large parts of Australia. By integrating 278 new sediment-derived Sr isotope measurements with global datasets of plants, soils, and

waters, and using a random forest regression model, we produced a continuous spatial prediction of Sr isotope variation at the continental scale.

The isoscape captures major geological controls, with higher $^{87}Sr/^{86}Sr$ values associated with ancient terrains such as the Yilgarn Craton and Palaeozoic provinces of eastern Australia, and lower values over younger sedimentary basins and coastal regions. Variable selection analysis identified bedrock $^{87}Sr/^{86}Sr$ values, geological age, dust/seaspray source distance, terrane age, and precipitation as key predictors. An associated standard deviation map highlights areas of higher uncertainty, particularly along coastal margins and in regions with complex geology.

Comparison with previous regional and global Sr isoscapes confirms that our model reliably reproduces broad isotopic patterns while providing significantly improved coverage for Australia. Although limitations remain due to sampling density and the use of sedimentary proxies, the resulting isoscape offers an important new baseline for provenance research, palaeoecological reconstructions, and environmental applications. Future work should focus on expanding bioavailable sampling coverage and refining models at finer spatial scales.

**Author contributions.** AD provided technical guidance, resources, supervision, data curation, and led the analysis, visualisation, and manuscript writing. FD produced the Sr isotope data and assisted with data curation. CB contributed to the isoscape calculation and assisted with manuscript editing. PdC conceived the study, provided samples and funding, contributed to data curation, and assisted with manuscript editing.

**Competing interests.** The contact author has declared that none of the authors has any competing interests.

**Disclaimer.** N/A

**Data availability.** The bioavailable Sr isotope dataset is available from the Geoscience Australia e-Catalogue entry by Caritat et al. (2025a) on https://dx.doi.org/10.26186/150024. Raster files for the isoscape and its uncertainty layer are available as Supplementary Material accompanying this manuscript.

**Acknowledgments** The National Geochemical Survey of Australia (NGSA) was made possible through Commonwealth funding provided by the Onshore Energy Security Program and Geoscience Australia appropriation funding (http://www.ga.gov.au/ngsa). We gratefully acknowledge the collaboration of geoscience agencies from all Australian states and the Northern Territory. We thank all landowners and custodians—private, corporate, and traditional—for granting access to sampling sites. We also acknowledge the support of Geoscience Australia laboratory staff for their assistance with sample preparation.

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

**Tables**

Table 1. Environmental covariates used in the random forest model to predict bioavailable $^{87}Sr/^{86}Sr$ values.

| No. | Variable Name | Description |
|---|---|---|
| 1 | r.m1 | Global median of the bedrock $^{87}Sr/^{86}Sr$ isotope ratio model |
| 2 | r.srsrq1 | First quartile (Q1) of the $^{87}Sr/^{86}Sr$ isotope ratio model |
| 3 | r.srsrq3 | Third quartile (Q3) of the $^{87}Sr/^{86}Sr$ isotope ratio model |
| 4 | r.meanage_geol | Mean geological age of the bedrock (Myr) |
| 5 | r.minage_geol | Minimum geological age of the bedrock (Myr) |
| 6 | r.maxage_geol | Maximum geological age of the bedrock (Myr) |
| 7 | r.age | Global mean terrane (basement) age (Myr) |
| 8 | r.GUM | Global Unconsolidated Material (GUM) type |
| 9 | r.bouger | Bouguer gravity anomaly |
| 10 | r.elevation | Elevation above sea level (m) |
| 11 | r.mat | Mean annual temperature (°C) |
| 12 | r.map | Mean annual precipitation (mm/year) |
| 13 | r.ai | Aridity Index |
| 14 | r.pet | Potential evapotranspiration (mm/day) |
| 15 | r.dust | Dust deposition rate (g/m²/year) |
| 16 | r.salt | Sea salt aerosol deposition (g/m²/year) |
| 17 | r.distance | Distance to reference points (e.g. rivers or coasts) |
| 18 | r.volc | Volcanic sulfur deposition (kg/m²/s) |
| 19 | r.fire | Black carbon from wildfires (kg/m²/s) |
| 20 | r.foss | Black carbon from fossil fuels (kg/m²/s) |
| 21 | r.clay | Soil clay content (%) |
| 22 | r.ph | Soil pH (converted to decimal) |
| 23 | r.cec | Cation exchange capacity (cmol$^+$/kg) |
| 24 | r.bulk | Soil bulk density (g/cm³) |
| 25 | r.ocs | Organic carbon stock (kg/m²) |

| No. | Variable Name | Description |
|-----|---------------|-------------|
| 26 | r.sw | Soil water content |

References for each variable raster are listed in Bataille et al. (2020)

**Table 2. Summary of statistics of this study's 278 bioavailable $^{87}Sr/^{86}Sr$ ratios by region.**

| Region | N | Min | Max | Mean |
|--------|-----|---------|---------|---------|
| N Aus | 152 | 0.70501 | 0.78121 | 0.72150 |
| SE Aus | 38 | 0.70739 | 0.71908 | 0.71277 |
| SW Aus | 88 | 0.71153 | 0.75274 | 0.72468 |

Values represent the number of samples (N), minimum, maximum, and mean $^{87}Sr/^{86}Sr$ ratios for each region.

**Figures**

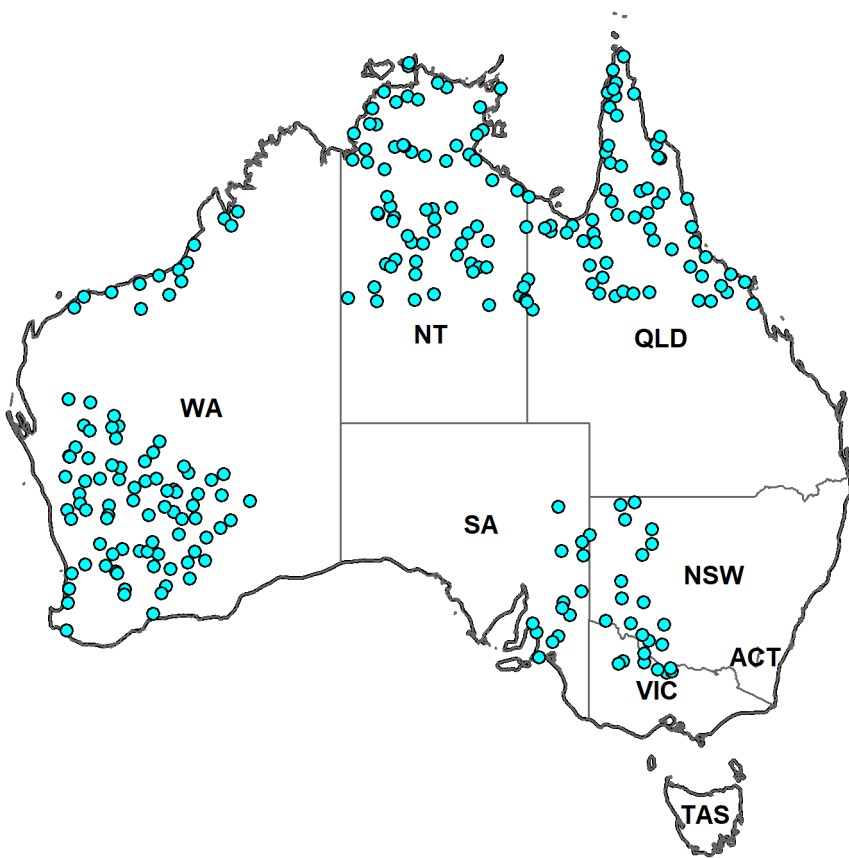

Figure 1. Map showing the locations of sediment samples analysed for bioavailable strontium (Sr) isotopes in this study (circles). Sampling sites span inland southeastern Australia (South Australia - SA, New South Wales - NSW, Victoria - VIC), northern Western Australia (WA), the Northern Territory (NT), Queensland (QLD) (north of 21.5°S), and the Yilgarn Craton in southern Western Australia.

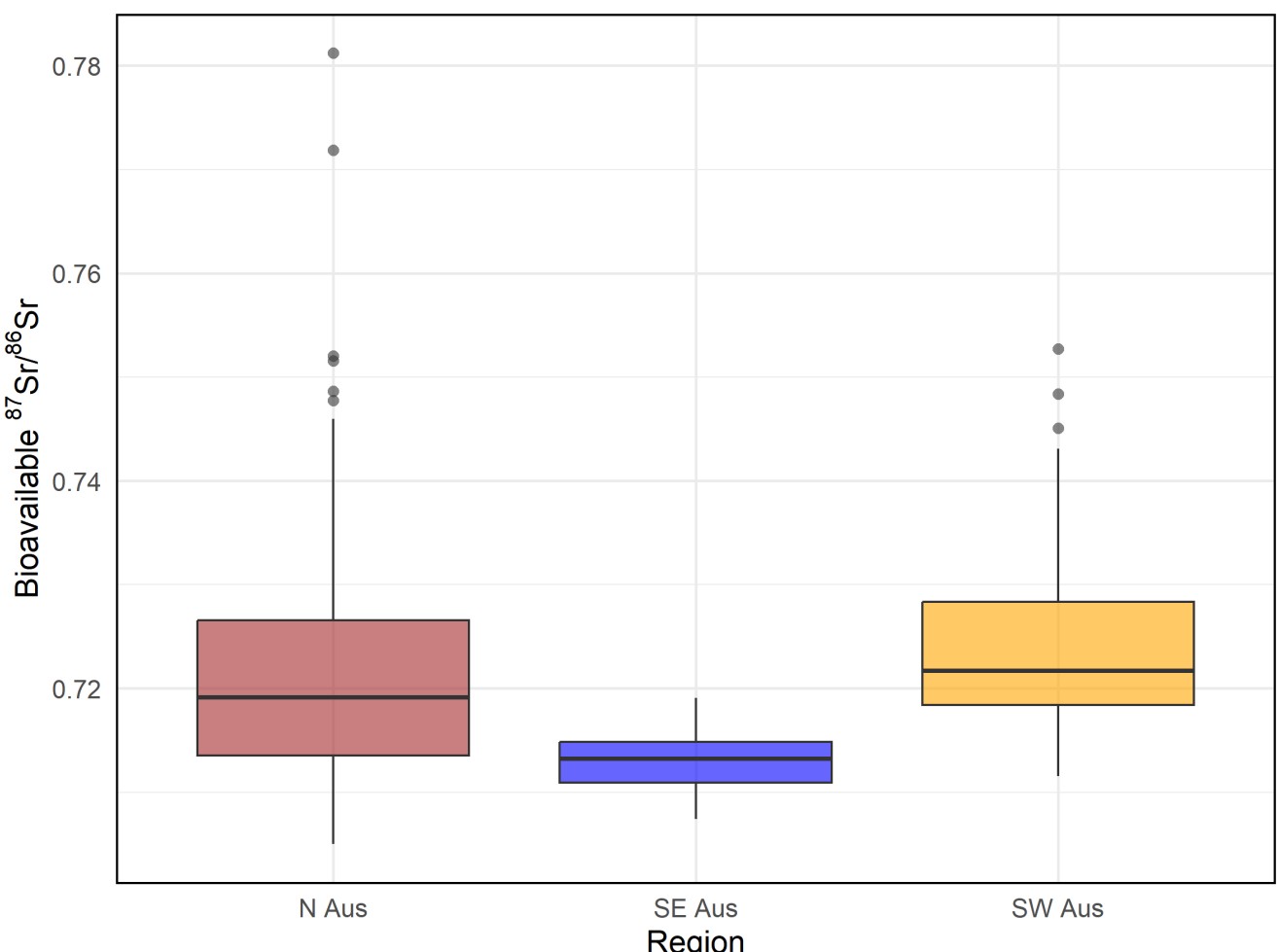

Figure 2. Boxplot showing regional variation in bioavailable $^{87}Sr/^{86}Sr$ ratios across northern Australia (N Aus), southeastern Australia (SE Aus), and southwestern Australia (SW Aus). The data highlight systematic regional differences, with N Aus having the widest range of values, SE Aus displaying the lowest mean $^{87}Sr/^{86}Sr$ values, and SW Aus showing overall more radiogenic signatures (highest mean).

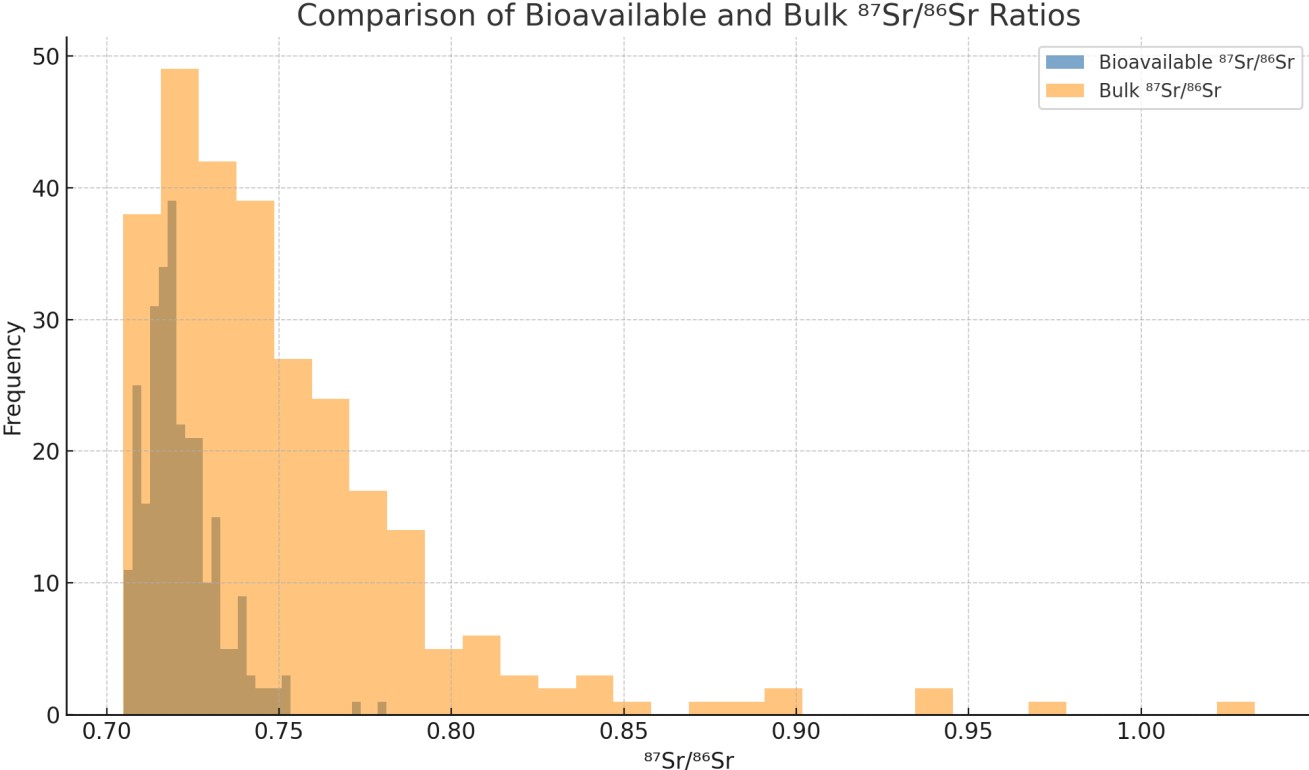

**Figure 3. Histogram comparing bioavailable (this study) and bulk $^{87}Sr/^{86}Sr$ ratios (Caritat et al., 2022, 2023, 2025b) across all samples. Bioavailable Sr (blue/grey; $n = 278$) displays a narrower and less radiogenic range, while bulk Sr (orange; $n = 576$) shows greater variability and extends to more radiogenic values. The broader range of bulk $^{87}Sr/^{86}Sr$ values reflects the inclusion of highly radiogenic minerals such as feldspars and micas within the bulk sediment, rather than being an artefact of the larger sample size compared to the bioavailable fraction.**

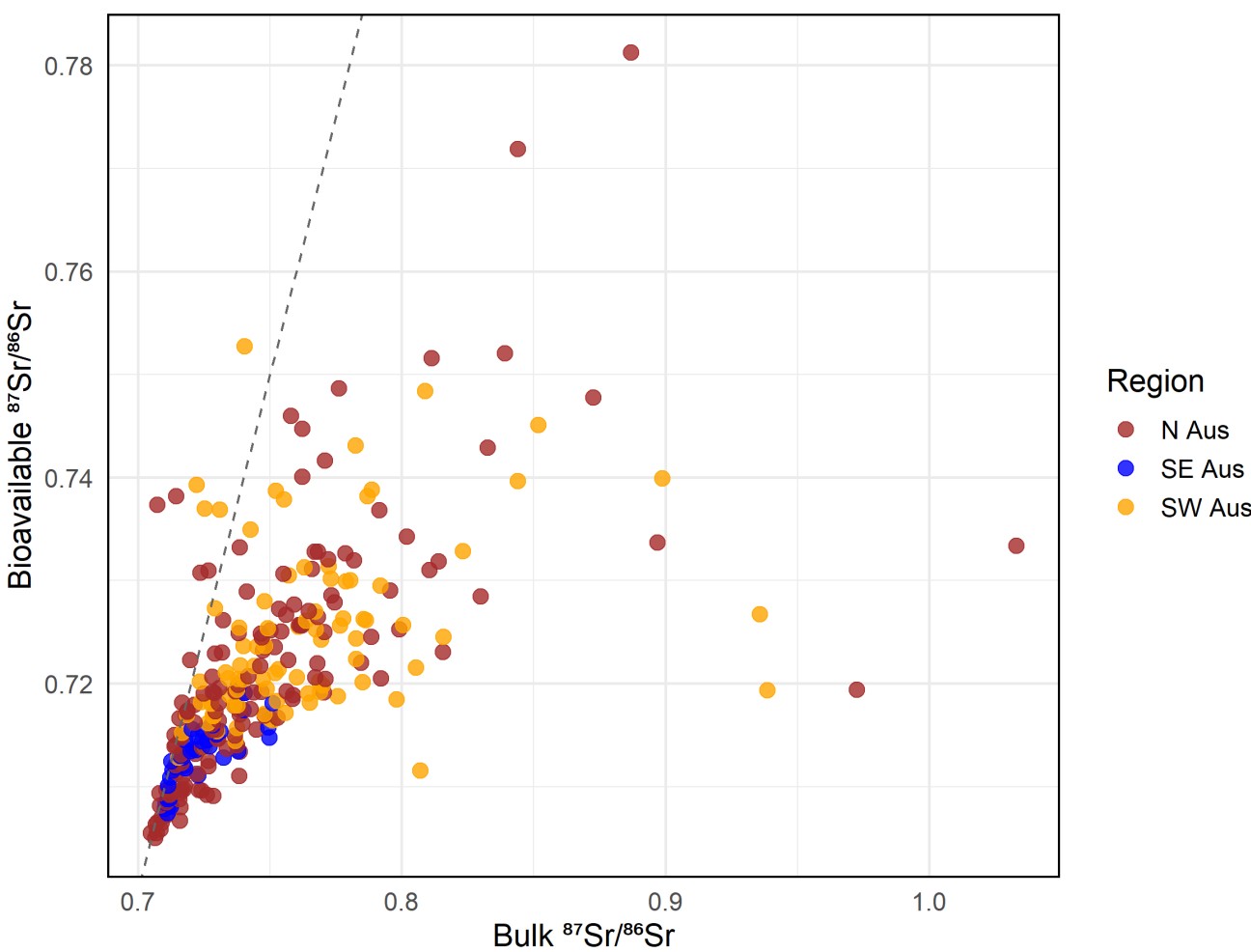

**Figure 4. Relationship between bioavailable and bulk $^{87}$Sr/$^{86}$Sr ratios for sediment samples across Australia. Points are coloured by region. The dashed line indicates a 1:1 relationship.**

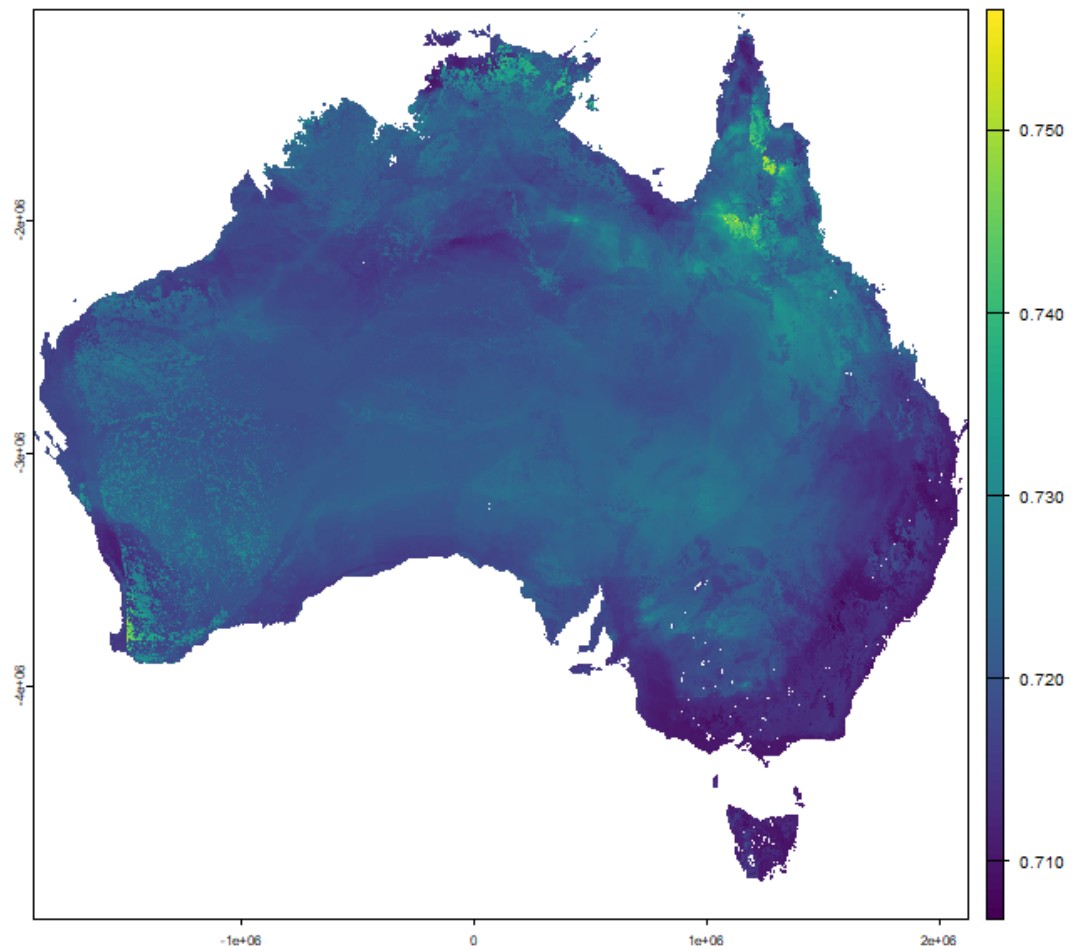

**Figure 5. Predicted bioavailable $^{87}$Sr/$^{86}$Sr isoscape of Australia generated using a random forest regression model trained on 278 samples from this study and global reference datasets. The colour scale represents predicted bioavailable $^{87}$Sr/$^{86}$Sr ratios, with lower values shown in purple and higher values in yellow. Higher predicted ratios are observed over ancient geological regions such as the Yilgarn Craton in Western Australia and parts of eastern Palaeozoic terranes. Lower values dominate younger sedimentary basins and coastal zones, reflecting geological age, regolith development, and potential marine or aeolian influences. An alternative version of this map with enhanced contrast in the lower value range (0.709–0.715) is provided in the Supplementary Material.**

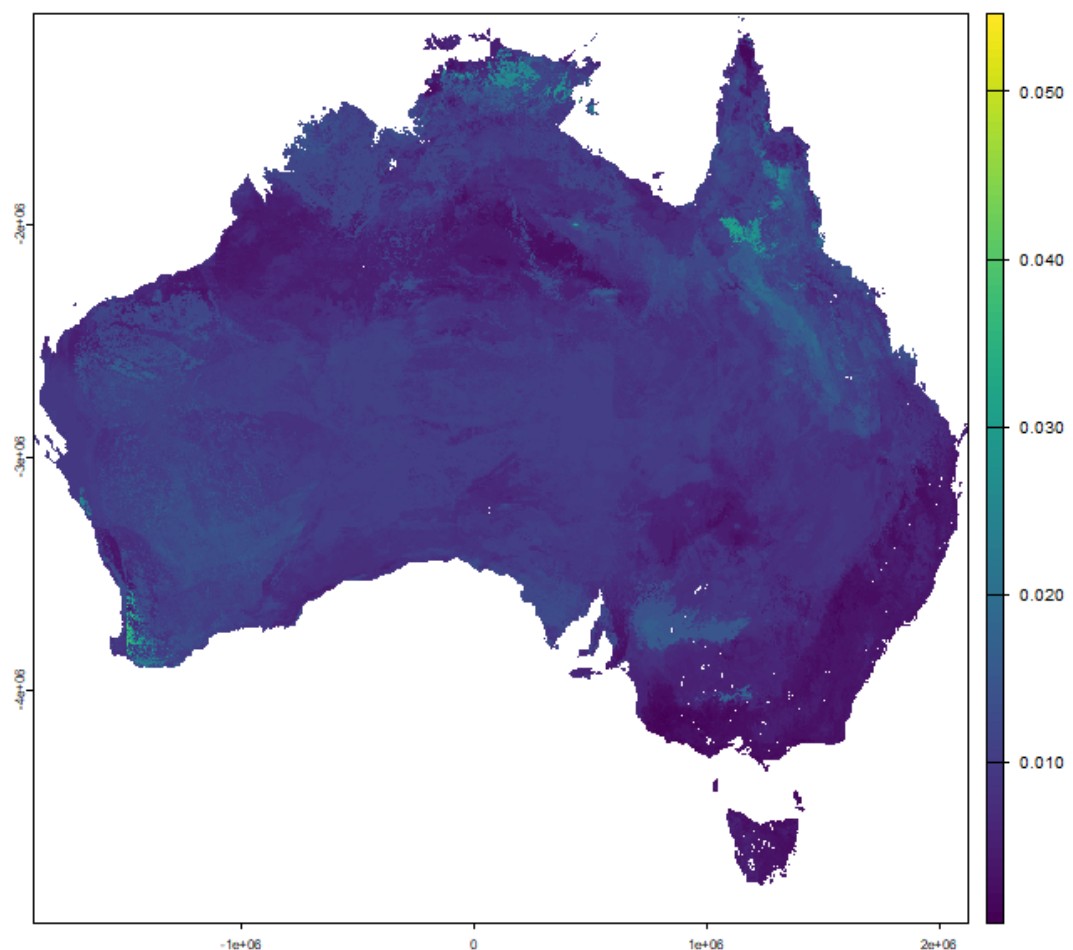

**Figure 6. Map of the predicted uncertainty (standard deviation, SD) in bioavailable strontium isotope ratios ($^{87}$Sr/$^{86}$Sr) across Australia. Uncertainty values reflect the spatial prediction error estimated from the random forest model, based on input variables such as geology, precipitation, and dust sources. Higher SD values indicate lower confidence in the predicted $^{87}$Sr/$^{86}$Sr ratios. An alternative version of this map with enhanced visual contrast is provided as Supplementary Figure S2.**

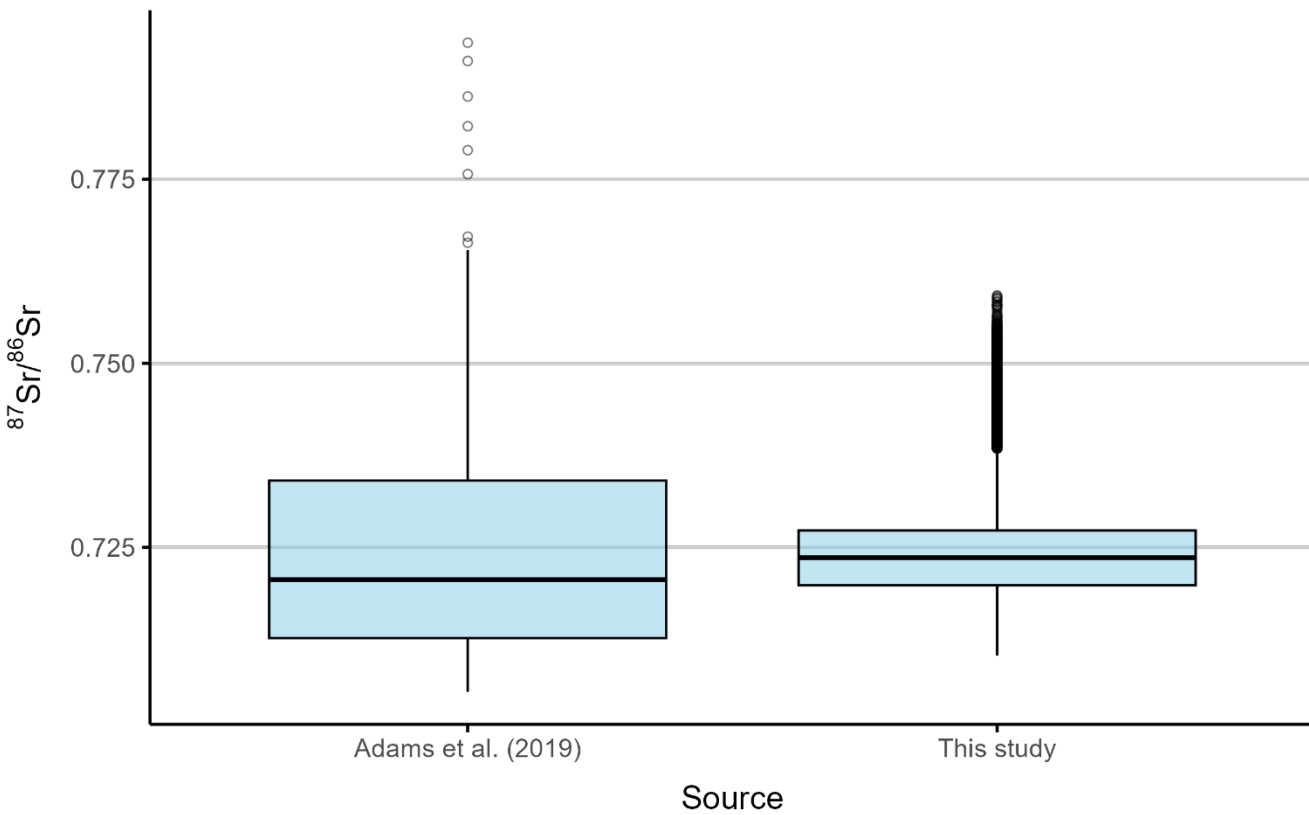

**Figure 7. Comparison of bioavailable $^{87}Sr/^{86}Sr$ ratios for the Cape York Peninsula between measured values reported by Adams et al. (2019) and predicted values from this study. The measured values are based on bioavailable Sr isotope ratios from plants, soils, and waters, while the predicted values were generated using a machine learning model trained on sediment samples and global reference datasets.** ~~Predicted values are on average 0.00124 lower than measured values, though this difference is not statistically significant (p = 0.25). Overall, the predictions capture the same isotopic range and spatial trends across Cape York.~~

**Supplementary Material**

Table S1. Breakdown of total dataset used in the isoscape model.

| Category | Count | Percent |
|---|---|---|
| Plant | 5152 | 23.6 % |
| Soil | 3511 | 16.1 % |
| Water | 5610 | 25.7 % |
| Other (animal, rock, etc.) | 7536 | 34.6 % |

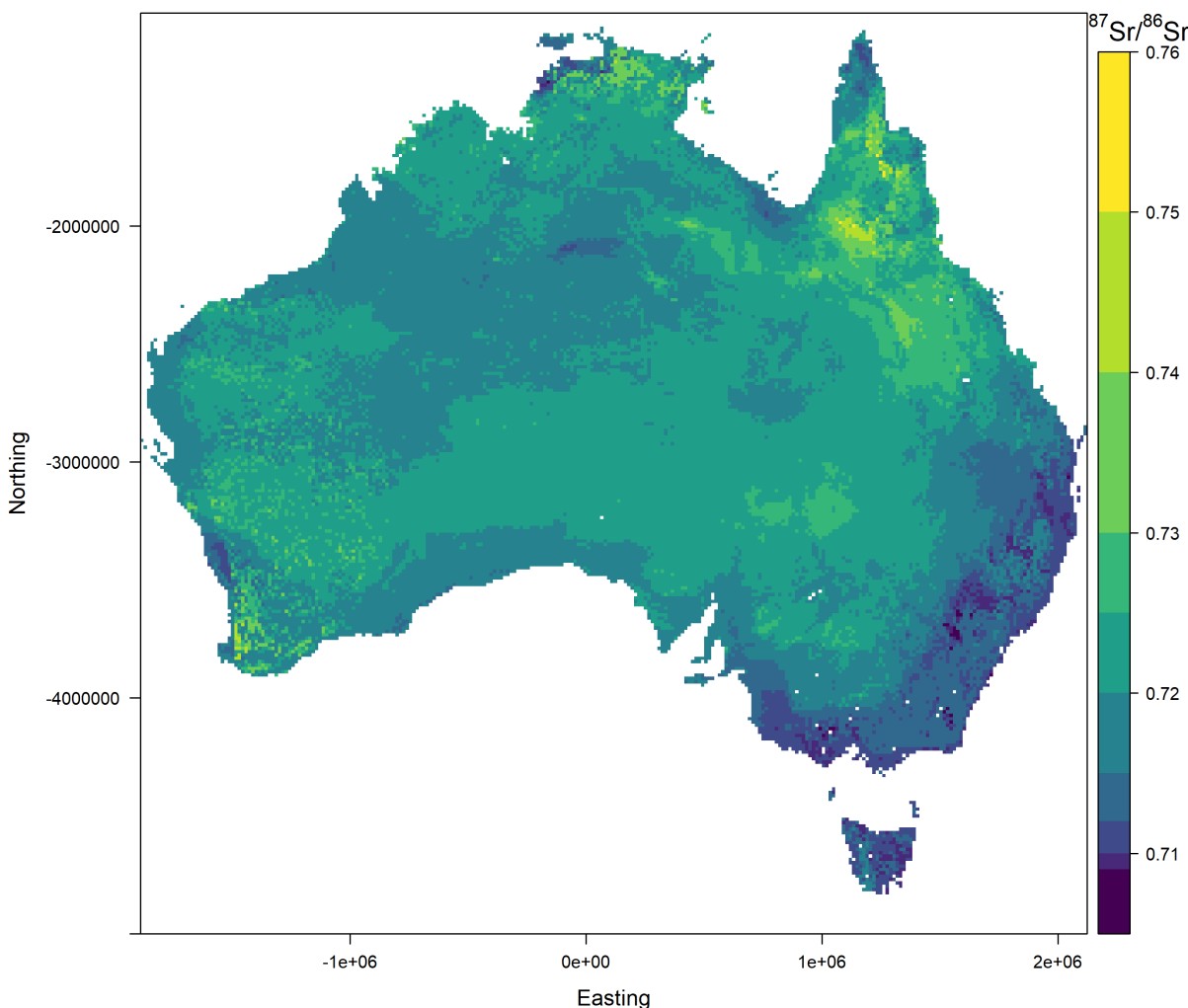

Supplementary Figure S1. Alternative version of the predicted bioavailable $^{87}Sr/^{86}Sr$ isoscape of Australia using a modified
colour ramp to enhance visual contrast in the lower range of values (0.709–0.715). This version highlights subtle isotopic
variation across northern and northwestern Australia that may be difficult to discern in the main figure.

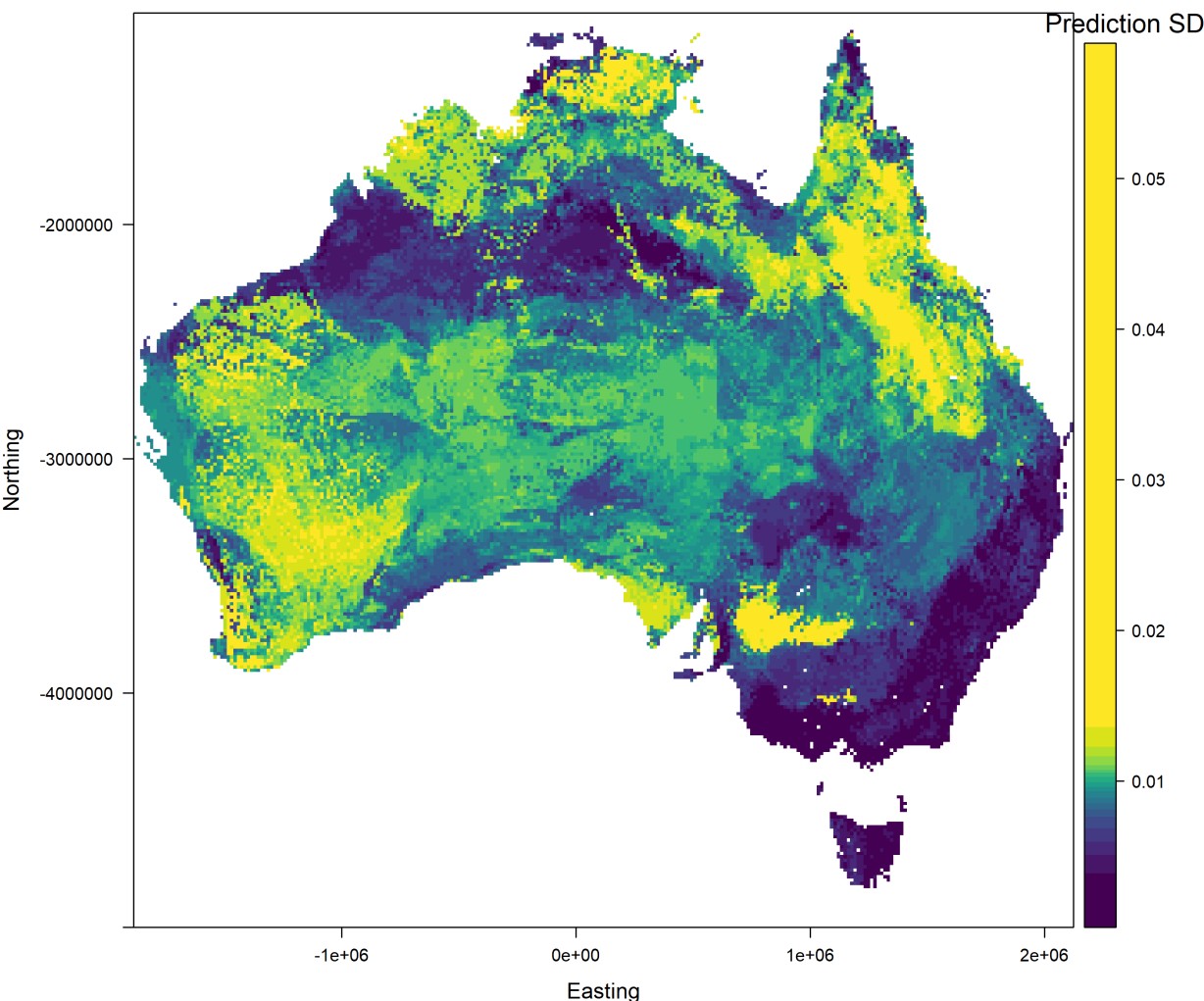

**Supplementary Figure S2.** Predicted uncertainty (standard deviation) in bioavailable $^{87}Sr/^{86}Sr$ values across Australia, based on the random forest model. This map uses a modified colour scale with non-linear breaks to improve visibility of spatial variability in low-standard-deviation areas. Elevated uncertainty occurs across parts of northern, southwestern, and central Australia, reflecting greater geological and environmental heterogeneity and/or sparse training data. In contrast, lower uncertainty is seen in more geologically uniform and well-constrained regions, including southeastern Australia