# Peer review of "A Bioavailable Strontium Isoscape of Australia"

_Earth System Science Data, 2025_

## Author Response (AR1)

**Response to Anonymous Referee #1**

**General**
We thank the reviewer for their constructive comments, which have helped improve the clarity and impact of the manuscript. Below we address each point in turn.
* * *
**Abstract**

**Comment:** *Can you better define the difference between the catchment outlet sediment samples used in this study, and the co-located bulk sediments to which results are compared.*

**Response:**
We have clarified this in the *Material and Methods* sections. The bioavailable Sr was extracted from the <2 mm fraction of 'top outlet sediment' (TOS) samples using $NH_4OAc$ leaching, whereas the bulk Sr values refer to total acid digestion of co-located sediments, inclusive of all mineral phases. This distinction is now explicitly stated in the revised manuscript.
* * *
**Introduction**

**Comment (Lines 49–50):** *Also Sr content and more specifically the elemental Rb/Sr ratio. A high Rb content will move the 87Sr/86Sr ratio little if the Sr content is very high…*

**Response:**
We agree and have revised the relevant passage to clarify that the Rb/Sr ratio, rather than Rb content alone, governs the radiogenic evolution of $^{87}Sr/^{86}Sr$. This change improves the accuracy of our geochemical context.
* * *
**Study Area / Material Section**

**Comment:** *Some repetition with Study Area section, these could be combined?*

**Response:**
We acknowledge the overlap between the *Study Area* and *Material* sections and have revised both to reduce redundancy while maintaining their distinct roles.
* * *
**Methods – Referencing SRM987**

**Comment:** *The $^{87}Sr/^{86}Sr$ data were not referenced to a specific value for SRM987…*

**Response:**
We now report the average $^{87}Sr/^{86}Sr$ value measured for SRM987 during the analytical campaign: 0.709810 ± 0.000044 (2SE, n = 90). As this is lower than the accepted value of 0.710252 ± 0.000013 from Weis et al. (2006), all sample $^{87}Sr/^{86}Sr$ ratios were normalised to the Weis et al. reference value using session-specific SRM987 means. This correction facilitates accurate inter-laboratory comparison and allows future users to re-reference the data if needed.
* * *
**Results – Fig 3 and Fig 4 Colour Consistency**

**Comment:** *Colours of histogram differ from legend*
**Comment:** *Use same colours for regions as in Fig 2…*

**Response:**
Figure 3 compares bioavailable and bulk $^{87}Sr/^{86}Sr$ values rather than regional subsets, so regional colour consistency does not apply. However, we have corrected the legend to ensure colours match the plotted data. In Figure 4, regional colours have been updated to match those used in Figure 2 for consistency across region-based plots.
* * *
**Fig 5 – Colour Range**

**Comment:** *The colour range used in Fig 5 makes it very difficult to discern subtle differences…*

**Response:**
We acknowledge that the colour scale in Figure 5 may limit visual differentiation of lower $^{87}Sr/^{86}Sr$ values, particularly in northern and northwestern Australia. To address this, we have added an alternative version of the isoscape to the Supplementary Material using a different colour ramp optimised to distinguish values in the 0.709–0.715 range.
* * *
**Fig 6 – Uncertainty Colour Scheme**

**Comment:** *…most of the figure is variable shades of dark blue.*

**Response:**
We acknowledge the reviewer's concern that most of Figure 6 appears as variable shades of dark blue, making it difficult to interpret spatial patterns in uncertainty. To address this, we have retained the main figure for consistency with the other maps, but included an alternative version in the Supplementary Material (Figure S2) that uses a modified colour ramp to enhance visual contrast in the lower SD range. This supplementary figure allows better differentiation of uncertainty across the full prediction surface.
* * *
**Discussion – Line 235**

**Comment:** *…Rb/Sr elemental ratio trumps Rb concentration only.*

**Response:**
Agreed. The discussion has been updated to state that the Rb/Sr ratio is the key control on radiogenic evolution of $^{87}Sr/^{86}Sr$, not absolute Rb concentration alone.
* * *
**Discussion – Line 245 (Fig 6 usability)**

**Comment:** *Can alternative schemes be tested?*

**Response:**
Yes. While the main figure has been retained for consistency with the rest of the manuscript, we have included an alternative version in the Supplementary Material (Figure S2) that uses a

modified colour scheme to enhance contrast across the range of prediction uncertainty. This version improves interpretability, particularly in areas with low standard deviation values.

**Conclusions / General Comment**

**Comment:** *The Sr isoscape produced is impressive... but a small test application would add greatly...*

**Response:**
We agree and have included a short application example in the *Discussion* section comparing predicted values to published plant/water/soil data from Adams et al. (2019) in Cape York Peninsula. This demonstrates the model's applicability to provenance studies and strengthens the practical relevance of the isoscape.

**Response to Referee #2 – Dr Ian Moffat**

**General**
We appreciate the reviewer's positive feedback and helpful suggestions. Below we respond to each concern raised.

**Introduction – Citations**

**Comment:** *...fail to offer any Australian references to support these statements...*

**Response:**
Thank you for this helpful suggestion — we were not previously aware of the study by Rippon et al. (2020), and have now included it. While the opening paragraph of the Introduction is intended to broadly frame the relevance of provenance tools across disciplines (without reference to specific isotopic systems), we have added citations to relevant Australian work at the point where strontium isotopes are first introduced. Specifically, we now cite Adams et al. (2019), Rippon et al. (2020), and Caritat et al. (2022, 2023, 2025b) to acknowledge existing contributions to Sr isotope provenancing and geochemical mapping in the Australian context.

**Main Concern – Use of Catchment Outlet Samples**

**Comment:** *...rather than those associated with the local bedrock... how might this have impacted results?*

**Response:**
We have expanded the *Discussion* to explain this methodological choice and its implications. Catchment outlet sediments were chosen for their integrative properties at regional scales. While this limits fine-scale resolution, they provide a robust first-order approximation of regional bioavailable Sr, suitable for broad-scale isoscape modelling. We acknowledge that point-sourced biological samples (e.g. plants) would be ideal for local-scale provenancing and encourage future work to incorporate such samples, especially for applications requiring finer spatial resolution.

**Additional Citation – Rippon et al. 2020**

**Comment:** ...*should probably also mention Rippon et al. 2020...*

**Response:**
Thank you for bringing this study to our attention. We have now cited Rippon et al. (2020) in both the Introduction and Discussion. In the Introduction, it is referenced alongside Adams et al. (2019) as an example of Australian Sr isotope data relevant to provenance studies. In the Discussion, we highlight Rippon et al. (2020) as an important regional effort to characterise bioavailable Sr in the Adelaide region using low-mobility fauna, further supporting the value of baseline Sr data for archaeological applications.

**Response to Anonymous Referee #3**

**Reviewer comment:** Lines 30–39 lacks citations.
**Response:** We thank the reviewer for this observation. We have now added the relevant citations in this section to support the statements made.

**Reviewer comment:** There should be a brief discussion in the paper that highlights the comparability of each sample type (plant, soil, water) to animal/human tissues.
**Response:** We agree that this is an important point. We have added a short discussion outlining the comparability of plant, soil, and water samples to animal/human tissues, and the considerations when using each of these sample types in provenance studies.

**Reviewer comment:** Furthermore, a detailed breakdown of the total sample used to generate the isoscape, particularly the percent plant, soil, and water samples used. This information could easily be included in the supplemental information.
**Response:** Thank you for this suggestion. We have now included a detailed breakdown of the total dataset, specifying the proportion of plant, soil, and water samples used, in the Supplementary Information (Table S1). Specifically, the database consists of 23.6% plant samples, 16.1% soil samples, and 25.7% water samples, with the remaining 34.6% representing other categories (e.g., animal tissues, shells, rock).

**Reviewer comment:** It is nice to see that the tiffs are included in the supplement.
**Response:** We appreciate the reviewer's positive feedback and are glad that the inclusion of the raster files is useful.